# Pre-COVID-19 Organic Market in the European Union—Focus on the Czech, German, and Slovak Markets

Stanislav Rojík [1,*], Martina Zámková [2], Martina Chalupová [3], Ladislav Pilař [1], Martin Prokop [2], Radek Stolín [2], Karel Malec [4], Seth Nana Kwame Appiah-Kubi [4], Mansoor Maitah [4], Paweł Dziekański [5] and Piotr Prus [6]

1   Department of Management, Faculty of Economics and Management, Czech University of Life Sciences Prague, Kamycka 129, 16500 Prague, Czech Republic; pilarl@pef.czu.cz
2   Department of Mathematics, College of Polytechnics Jihlava, Tolstého 16, 58601 Jihlava, Czech Republic; martina.zamkova@vspj.cz (M.Z.); martin.prokop@vspj.cz (M.P.); radek.stolin@vspj.cz (R.S.)
3   Department of Economics Studies, College of Polytechnics Jihlava, Tolstého 16, 58601 Jihlava, Czech Republic; martina.chalupova@vspj.cz
4   Department of Economics, Faculty of Economics and Management, Czech University of Life Sciences in Prague, 16500 Prague, Czech Republic; maleck@pef.czu.cz (K.M.); appiah-kubi@pef.czu.cz (S.N.K.A.-K.); maitah@pef.czu.cz (M.M.)
5   Department of Economics and Finance, Jan Kochanowski University in Kielce, Uniwersytecka 15 St., 25-406 Kielce, Poland; pawel.dziekanski@ujk.edu.pl
6   Laboratory of Economics and Agribusiness Advisory, Department of Agronomy, Faculty of Agriculture and Biotechnology, Bydgoszcz University of Science and Technology, 430 Fordońska St., 85-790 Bydgoszcz, Poland; piotr.prus@pbs.edu.pl
*   Correspondence: rojiks@pef.czu.cz; Tel.: +420-774-350-979

**Abstract:** This article compares attitudes to buying organic food in selected countries in Central Europe. The current research was conducted in 2019 on a total sample of 2800 respondents in the Czech Republic, Slovakia (Central Europe, with a relatively new organic food market), and Germany (a traditional Western Europe country with a mature food market). The study results demonstrate significant differences between the three selected markets. The product quality is the most important for German consumers. Slovak consumers consider organic food to be the least recognizable and least promoted of all the regions surveyed, and they are also the least likely to encounter targeted advertising for organic products. Germany is the country where most respondents regularly or occasionally buy organic food. In Slovakia, an interesting finding is the highest proportion of respondents who do not buy organic food at all. Czech respondents often buy organic products in specialized shops and like to grow organic products themselves. The results also suggest that Slovak consumers slightly more often prefer foreign organic products to the local ones, whereas consumers in Germany select regional organic products more often and prefer to buy regional products at farmers' markets.

**Keywords:** consumer behaviour; organic products; foreign; domestic and regional production; Czech Republic; Slovakia; Germany; correspondence analysis

## 1. Introduction

Consumers' constantly growing interest in organic food has caught strong attention of the academic community, as there is increasing demand for research on profile, habits, and attitudes of consumers towards the topic. Globally, the organic food market grew the most around the turn of the millennium (by about 80%) [1]. In 2018, i.e., still in pre-COVID-19 times, the organic food market was clearly dominated by the US (42%) followed by European countries lead by Germany and France with 11% and 9% of the global market share, respectively [2]. According to Sahota [3] organic food and drink sales amounted to almost EUR 107 billion in 2019. The world's most important market (the biggest single consumer market), the United States, reached more than EUR 44 billion at the same time,

with Germany remaining the leading market for organic products in Europe with an 11.4% share of global organic sales [4]. Sahota [3] pointed out that even though global organic food sales are growing at a significant pace, there are still persistent challenges. One of them is the fact that demand for the organic food production is concentrated in Europe and North America.

The action plan of the EU's Common Agricultural Policy complements the new legal framework for organic agriculture (Regulation (EU) 2018/848). The plan aims to stimulate both supply and demand of organic products, focusing to build more profound trust of consumers through promotion activities and green public procurement [5].

The interest in organic and domestic (local) food products and the preference for them have steadily risen in the EU over the past two decades. For food producers and marketers, it is crucial to be aware of the changing determinants of organic and local food purchasing in pre-COVID-19 times and during the COVID-19 pandemic [6–11]. It is also of interest to food policymakers for multiple reasons, including the fact that organic food production is based on the use of sustainable technologies with a positive impact on ecological systems and biodiversity [12,13]. An extensive number of articles published around the globe identified constructs to help build a thorough understanding of factors and motivators driving consumers' behaviour in choosing the organic and regional or local food products. It is considered to be a dynamic, multifaceted, and contextual phenomenon [14] influenced by various factors, such as moral norms [15,16], subjective factors, such as health consciousness and wellbeing, or hygienical reasons [17–21], social pressures [22–24], and geographical distance [25–27], among other things.

The aim of this research is to compare customer preferences regarding organic food in three EU countries: the Czech Republic, Germany, and Slovakia. The territories of the Czech Republic and Slovakia were part of one country for 75 years, and therefore share close business ties, including similar development traits. Germany was chosen because of its geographical and cultural proximity and as a benchmark, the country with the largest organic food market in Europe, and because the Czech Republic and Slovakia have the strongest economic ties with Germany. The data presented monitor the buying behaviour and preferences of consumers before the COVID-19 pandemic, which has had a tremendous impact on many aspects, and thus has probably provided the organic market an unprecedented upturn in many countries.

## 2. Organic Food Market

One of the dominant theoretical approaches to guide research on intentional behaviour, explaining the impact of attitudes, is Ajzen's Theory of Planned Behaviour [28] that examines the strength of the relationships between attitude, subjective norms, perceived behavioural control, and intention, as well as between intention and behaviour. The theory has often been applied in the area of food choice and is also extensively used to explain and predict agriculture, farming and food (including organic) consumption [29–34]. With further theoretical development, such as in the field of self-regulation theories [35], theories with emphasis on temporal dynamics and temporal frames, such as temporal self-regulatory theory [36] explaining how individuals consider benefits and costs of behavioural options [37–39] or health action process approach (HAPA) as a theoretical framework to understand behaviour change [40], these theories help explain and understand deeper behavioural phenomena in connection with organic food choice.

When it comes to examining consumer behaviour related to organic food, what matters are the consumers' attitudes and reasons for purchasing products labelled as such. For instance, Pilař et al. [41] identified two main reasons why consumers purchase these products: egoistic and altruistic motivations. The main reasons for the egoistic motivation include factors, such as higher perceived quality of organic food and belief that organic food is associated with health benefits and better nutritional characteristics compared with conventional products. The altruistic motivation is mainly based on satisfying higher level needs, such as solidarity with a specific region, ecological reasons, or animal welfare [42,43].

Research focusing on the German market suggests that consumers' preference of organic product is driven by their sustainability concerns, together with preference for the naturalness and healthiness of such products [44,45]. Zagata [46] who studied the Czech market suggested that the consumers perceive organic products as 'food without chemicals', with favourable effects on health, which was confirmed by the research of Soroka, Mazurek-Kusiak, and Trafialek [47]. In this research, the authors also suggest that both Slovaks and Czechs value highly the effects of organic food on health, Slovak respondents saw as a critical barrier a short expiry time of such products.

Other significant determinants on the German market were the preference for local and domestic food with higher quality [23,44,48], which resonates on the Czech and Slovak markets as well [49,50]. The most common negative effect on the purchase intentions in Germany (as well as in the Czech Republic and Slovakia) is the price consciousness, accompanied by an unfavourable economic situation [51–54]. Živělováe and Crhová [55] confirmed the growing interest of Czech consumers in these products despite a greater price difference compared with conventional food products than in Western Europe.

However, there are still many fundamental questions to be answered about the price-related purchasing behaviour of consumers. Hamm, Aschemann, and Riefer [56] argue that communicating a justification of high prices for organic food should be less important than using pricing and communication measures suitable for diverting consumers' attention from the absolute price of organic food and improving consumers' perception of the price-performance relationship in organic food. On all the three markets, there are obviously consumer segments that demonstrate strong organic values and attitudes, have sufficient income, and are willing to pay the premium price which will be viewed as a signal of the desired high quality [57]. Aschemann-Witzel and Zielke [58] and Cristache et al. [59] argue that price sensitivity depends on the consumer segment and product category, and organic consumers' price sensitivity tends to be lower than that of occasional or nonorganic buyers. Based on the literature review, the authors summed up common factors with substantial effects on the organic food markets in selected countries (Germany, the Czech Republic, and Slovakia) in the Table 1.

**Table 1.** Common factors' effect on the organic markets in selected countries.

| Factor | Markets | Research | Presumed Effect |
|---|---|---|---|
| Price sensitivity towards organic food | German | Aschemann-Witzel and Zielke (2017); Hamm, Aschemann, and Riefer (2007); Rödiger, Hamm (2015) | Negative |
| | Czech | Zámková and Prokop (2014) | Negative |
| | Slovak | Kádeková et al. (2017); Predanocyová et al. (2018) | Negative |
| Limited awareness of existing food labels and certifications | German | Wunderlich et al. (2018) | Negative |
| | Czech | Chalupová et al. (2016) | Negative |
| Presumed positive effect on health | German | Janssen (2018); Seegebarth et al. (2016) | Positive |
| | Czech and Slovak | Kádeková et al. (2017); Kozelová et al. (2013); Soroka et al. (2021) | Positive |

Source: (Authors).

Other important purchasing barriers for organic food include the lack of immediate availability, sensory criteria, daily purchase routines resulting in automatic and fast decisions, lack (overload) of information or low trust and limited awareness of existing food labels and certifications [60–63]. Scholars also focus their research on a frustrating paradox, the intention-behaviour gap or a green gap: terms that represents the inconsistency be-

tween consumers' declared concern about environmental problems and their actual buying behaviours [64–66]. However, these factors were not sufficiently investigated, especially on the Czech and Slovak markets, so it was not possible to compare the data with the situation on the German market.

*Organic Market in the EU with Focus on Germany, the Czech Republic, and Slovakia*

The European organic sector is developing dynamically, the consolidated data for 2019 that were published recently by FiBL show a continued growth for both the organic market and production. The most current data also demonstrate that organic farmland in the EU has reached about 14.6 million hectares, representing a share of 8.1% of all the farmland [3]. The largest area of organic farmland in 2019 in the EU was Spain (2.35 million hectares, representing a share of 9.7%), followed by France at 2.24 million hectares (7.7%). Germany crossed the 1 million mark (1.61 million hectares; 9.7%), the Czech Republic was the seventh in the EU with 0.54 million hectares (15.4%) and Slovakia had almost 0.2 million hectares (10.3%). By 2030, the European Commission is aiming to reach 25% of organic area [67].

The organic market grew dynamically in Germany, it almost doubled between 2010 and 2019, the Czech market almost tripled, while in Slovakia it remained the same. Germany represents the biggest organic market in Europe, and the world's second largest after the United States [3]. The retailers may play a crucial role in this development. According to Sahota [3], supermarkets played a driving force on the organic market, founding vital partnerships with organic associations, selling products with their labels, creating more opportunities for the farmers and local food producers.

In 2019 EU consumers spent EUR 84 on organic food per person, consumer spending on this market has doubled in the last decade. Such as in the last decade, the highest per capita consumption of organic food was in Denmark (EUR 344) and Switzerland (EUR 338) [3].

Germany is among the European countries with the highest per capita consumption of organic food (EUR 144 in 2019) [68]. In Central and Eastern European countries, consumer spending on organic food remains low; however, retail sales data are scarce for some countries, not being regularly updated [3]. The latest available data show, that even with the growth of the organic food consumption in the Czech Republic, it remains a niche market. In 2018 it accounted for just 1.58% in the total consumption of food and beverages [69], with EUR 16 spending on organic food per person in 2018 [3]. The same situation prevails in Slovakia, where the organic food represented only 0.83% in the total consumption in 2018, with consumers spending only EUR 1 on organic food per person in 2018 [3]. Main research questions and hypothesis of the article and our research were focused in following areas: How are available organic food products in the selected countries (Germany, Czech Republic, and Slovakia)? How recognizable are organic food products in the selected countries? How promoted are organic food products in the selected countries? Where are organic food products mostly bought in the selected countries? The place where respondents mostly buy organic products in the selected countries and the preference of local food in the selected countries.

## 3. Materials and Methods

The primary data were acquired through marketing research, a survey was distributed in several countries of the EU, in the Czech Republic, Slovakia, and Germany. A total of over 2800 respondents took part in the research. A representative (quota) sample of nearly 1000 respondents was contacted in each of the countries surveyed. The survey was conducted online in 2019 (March–June). Participants were contacted via an online questionnaire. Table 2 shows the distribution of respondents in individual countries.

**Table 2.** Relative distribution of respondents in individual countries.

| Country | Variables | Values | Percentage |
|---|---|---|---|
| Czech Republic | Sex | Women | 62.6% |
| | | Men | 37.4% |
| | Age | Less than 36 years | 64.9% |
| | | 36 years and more | 35.1% |
| | Education | Primary | 8.2% |
| | | High school | 70.1% |
| | | University | 21.7% |
| Slovak Republic | Sex | Women | 64.7% |
| | | Men | 35.3% |
| | Age | Less than 36 years | 61.9% |
| | | 36 years and more | 38.1% |
| | Education | Primary | 8.6% |
| | | High school | 74.1% |
| | | University | 17.3% |
| Germany | Sex | Women | 61.9% |
| | | Men | 38.1% |
| | Age | Less than 36 years | 64.5% |
| | | 36 years and more | 35.5% |
| | Education | Primary | 7.5% |
| | | High school | 67.3% |
| | | University | 25.2% |

Source: (Authors' calculations).

The questions in the questionnaire mainly focused on where the respondents purchase organic food (place of purchase), on the frequency of purchasing this type of food, the preferred origin of organic food (foreign countries, domestic production, and regional production), and the level (extent) of promotion of organic food in the different countries.

The data were analysed using the contingency tables analysis, including the Pearson's chi-square test, for more details, please see [70–74]. The correspondence analysis represents a popular graphical method often employed in order to analyse the associations between individual categories of one or more variables in contingency tables. The correspondence analysis mechanisms allow for the description of the associations between nominal or ordinal variables and their graphical presentation in multidimensional space [75]. The correspondence analysis describes the correlations of variables listed in contingency tables and deals with a similar problem as the principal component analysis which explains the correlations of initial continuous variables with a reduced number of latent components.

The correspondence analysis observes the correlations between individual categories of two categorical variables.

The results of the correspondence analysis are graphically displayed in correspondence maps, the axes of the reduced coordinate system, representing different categories of both variables. This analysis decomposes the chi-squared statistic associated with the table into orthogonal factors. There is a distance between single points which is described as a chi-squared distance. In order to express the distance between the $i$th and $i'$th row we used the formula:

$$D(i, i') = \sqrt{\sum_{j=1}^{c} \frac{\left(r_{ij} - r_{i'j}\right)^2}{c_j}} \tag{1}$$

where $r_{ij}$ represent the elements of row profiles matrix R and weights $c_j$ correspond to the elements of column loadings vector $c^T$. The objective of this analysis is to reduce the multidimensional space of both row and column profiles, plus to save maximally original data information, as suggested by Hebák et al. [76].

Where the response variable proves to be categorical, logistic regression was used. Explanatory variables may be continuous as well as categorical. In a binary logistic regression,

the response variable *Y* is dichotomous with the values of 1 and 0, indicating the presence or absence of an event A. Regression model parameters were estimated by the Maximum Likelihood Estimation. The Wald statistics tests the statistical significance of regression coefficients [77]. As for the primary data processing, the Statistica software proved to be very effective.

## 4. Results

The food quality represents the decisive factor mainly for the respondents from Germany, but the difference between the three groups is not that significant, since quality is viewed as important for over 95% of respondents. The situation is similar when it comes to the price, it is equally important for almost 90% of respondents.

Respondents from the Czech Republic (CR) find the appearance more important than the others, as seen in Table 3. Significant statistical dependence was proven, the *p*-value is less than 0.01.

**Table 3.** Contingency table, column relative frequencies: what is crucial for you when buying food, appearance and respondent's country of origin.

|  | Czech Republic | Slovak Republic | Germany |
|---|---|---|---|
| Definitely yes | 31.28% | 22.67% | 17.09% |
| Rather yes | 47.80% | 51.00% | 53.85% |
| Rather no | 16.65% | 19.67% | 22.22% |
| Definitely no | 2.57% | 4.67% | 4.27% |
| I do not know | 1.70% | 2.00% | 2.56% |

Source: (authors' calculations).

The brand of the product is considered to be important mostly by the Slovaks; respondents from Germany consider it the least important, as seen in Table 4. The significant statistical dependence was proven, the *p*-value is less than 0.001. The availability of products represents a crucial factor mainly for the Slovaks; however, the differences are rather insignificant, availability is important for approximately 70% of respondents. The place of origin is approached similarly in all countries; the Slovaks reported only slightly higher interest, approximately by 50% of respondents.

**Table 4.** Contingency table, column relative frequencies: what is crucial for you when buying food—brand and respondent's country of origin.

|  | Czech Republic | Slovak Republic | Germany |
|---|---|---|---|
| Definitely yes | 5.43% | 8.00% | 1.71% |
| Rather yes | 29.41% | 31.00% | 26.50% |
| Rather no | 49.42% | 44.33% | 45.30% |
| Definitely no | 12.72% | 14.33% | 25.64% |
| I do not know | 3.02% | 2.33% | 0.85% |

Source: (authors' calculations).

Approximately two thirds of respondents agreed with the support for organic farming coming from public sources, while Slovaks are the most accommodating. The Germans are apparently more confident about the health benefits of organic food, as about 80% of respondents claimed so. Organic food is considered to be tastier more often by the citizens of the Slovak Republic, as was confirmed by less than 50% of Slovak respondents. The Germans (less than 50% of respondents) tend to trust more often that organic food looks more appealing. Higher quality of organic food is perceived more often by respondents from the Germany (approximately 75% respondents).

The responses imply that organic produce is more accessible in the Germany, less accessible in the Czech Republic, and least accessible in Slovakia, as seen in Table 5. Significant statistical dependence was proven, *p*-value is less than 0.01. The same result is reflected in the correspondence map, see Figure 1.

**Table 5.** Contingency table, column relative frequencies: is organic food included in the assortment of your favourite grocery store? Respondent's country of origin.

|  | Czech Republic | Slovak Republic | Germany |
|---|---|---|---|
| Yes, it is | 67.00% | 62.39% | 69.64% |
| No, it is not | 11.00% | 16.24% | 7.79% |
| I have not searched for it in a grocery store | 16.00% | 16.24% | 19.06% |
| I do not know | 6.00% | 5.13% | 3.52% |

Source: (authors' calculations).

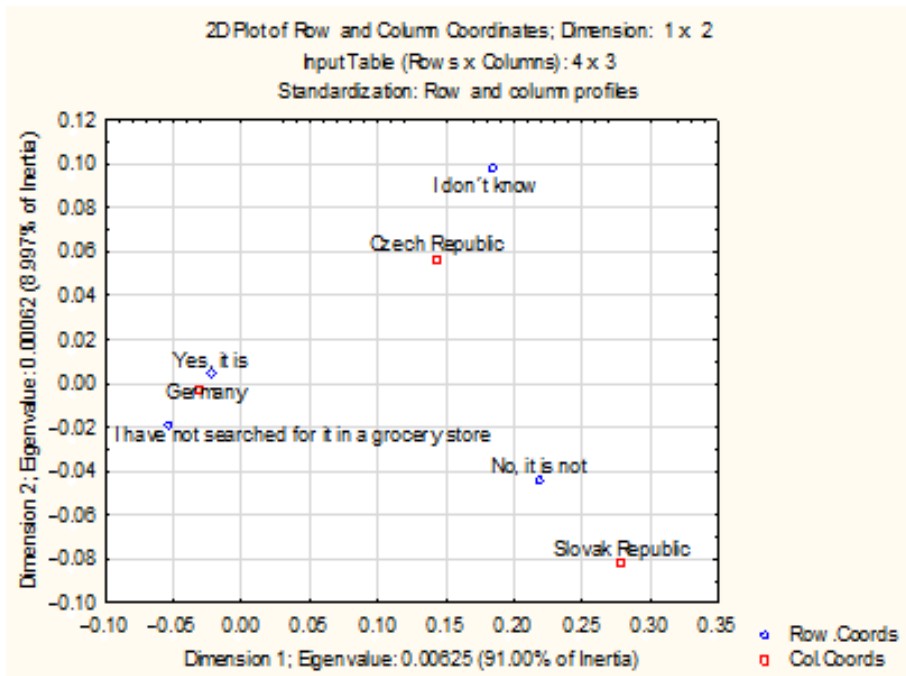

**Figure 1.** Correspondence map: is organic food included in the assortment of your favourite grocery store? Respondent's country of origin, Source: (authors' calculations).

Organic food is most easily recognizable in Germany, as opposite to Slovakia, where it is the least recognizable, as seen in Table 6. However, the dependency is not statistically significant. Significant statistical dependence was not proven, *p*-value is 0.16.

**Table 6.** Contingency table, column relative frequencies: are you convinced that organic food is sufficiently recognizable in your favourite grocery store? Respondent's country of origin.

|  | Czech Republic | Slovak Republic | Germany |
|---|---|---|---|
| Yes, it is | 51.41% | 47.33% | 54.70% |
| No, it is not | 24.03% | 30.33% | 21.37% |
| I do not know, I cannot consider | 24.57% | 22.33% | 23.93% |

Source: (authors' calculations).

Organic food is apparently poorly promoted in Slovakia, as seen in Table 7. The significant statistical dependence was proven, the *p*-value is less than 0.001. The correspondence map (Figure 2) below is an indication of the aforementioned weaker promotion of organic food in Slovakia.

**Table 7.** Contingency table, column relative frequencies: are you convinced that organic food is sufficiently promoted? Respondent's country of origin.

|  | Czech Republic | Slovak Republic | Germany |
|---|---|---|---|
| Definitely yes | 10.98% | 8.00% | 5.98% |
| Rather yes | 30.49% | 23.00% | 32.48% |
| Rather no | 43.54% | 45.67% | 40.17% |
| Definitely no | 8.53% | 15.00% | 8.55% |
| I do not know | 6.46% | 8.33% | 12.82% |

Source: (authors' calculations).

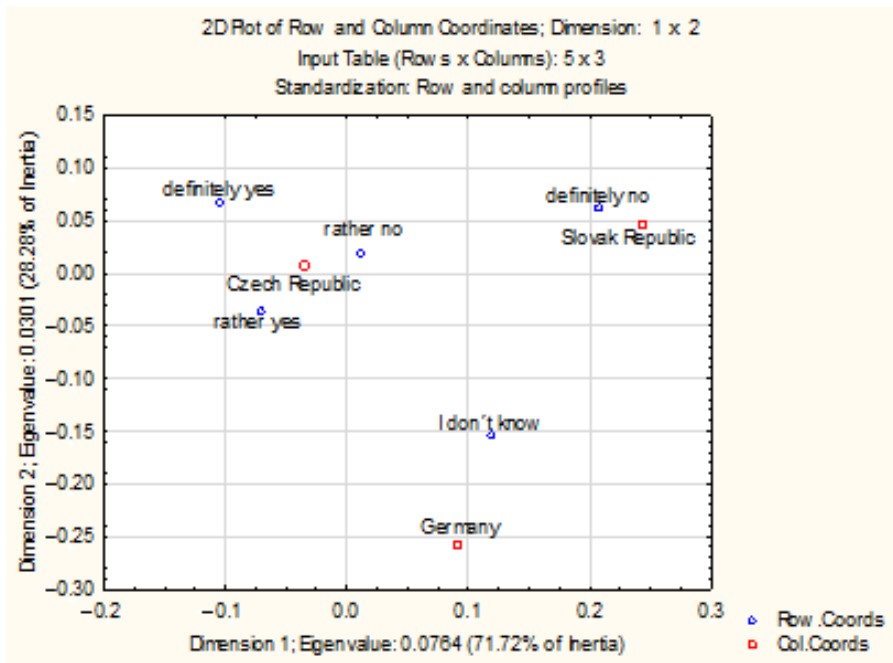

**Figure 2.** Correspondence map: are you convinced that organic food is sufficiently promoted? Respondent's country of origin, source: (authors' calculations).

As for the advertisements for organic food, the differences in respondents' answers are minimal; note that the largest proportion of respondents who have never seen any organic food ads come from Slovakia, as seen in Table 8. The significant statistical dependence was not proven, the *p*-value is 0.82.

**Table 8.** Contingency table, column relative frequencies: have you ever seen an advertising of organic food? Respondent's country of origin.

|  | Czech Republic | Slovak Republic | Germany |
|---|---|---|---|
| Yes, on billboards | 1.66% | 1.67% | 1.71% |
| Yes, in magazine, newspaper | 18.97% | 18.33% | 19.66% |
| Yes, on TV | 13.63% | 11.00% | 12.82% |
| Yes, on internet | 18.39% | 15.67% | 18.80% |
| Yes, I just do not know where it was | 23.03% | 26.00% | 19.66% |
| I do not know | 9.73% | 9.33% | 12.82% |
| No, I have not | 14.58% | 18.00% | 14.53% |

Source: (authors' calculations).

Germany is the place, where people tend to buy organic food most regularly or sometimes, while the Slovak Republic is at the other end of the scale, as seen in Table 9. The significant statistical dependence was proven, the *p*-value is less than 0.01.

**Table 9.** Contingency table, column relative frequencies: do you buy organic food in your household? Respondent's country of origin.

|  | Czech Republic | Slovak Republic | Germany |
|---|---|---|---|
| Yes, regularly | 8.67% | 6.26% | 11.97% |
| Yes, sometimes | 49.67% | 46.73% | 53.85% |
| I do not know, I do not follow whether it is organic food or not | 17.00% | 15.78% | 17.09% |
| No, never | 24.67% | 31.23% | 17.09% |

Source: (authors' calculations).

Some respondents indicated the reasons why they never purchase organic food. Having never thought about it was the most common reason in the Czech Republic, 14%, compared with approximately 10% everywhere else. The proportion of respondents, who do not trust the food is non-chemical is larger in the Czech and Slovak Republics (over 15%, as opposed to 10% in Germany), while the proportion of respondents who do not know where to buy organic food is minimal (around 1%, in Germany it is 6% of respondents). Citizens from the Czech Republic claim to be the least informed about organic food (8% versus 5% in the remaining countries). Approximately 13% of the respondents consider organic food to be unnecessary luxury (more often in Slovakia, least often in the Czech Republic). The difference is only about 1%. Approximately 7% of respondents reported the issue of insufficient assortment (over only 4% in Slovakia). The highest number of respondents who claim that organic food is not attractive enough come from the Czech Republic (almost 17%) and the lowest from Germany (almost 9%). Czech respondents appear to think that organic food is too expensive most often (30.5%) and the Germans think this least frequently (under 18%). In Central Europe, a rather insignificant number of respondents are not aware of this kind of food (about 0.7%), in Germany, it is 8%.

Big stores are more often frequented in Germany, least often in the Czech Republic, as seen in Table 10. Czech respondents tend to shop organic food in specialized stores, German respondents prefer to buy it at the farmers' markets. Only a minimum number of people in Germany grow organic produce themselves. The significant statistical dependence was proven, the *p*-value is less than 0.001.

**Table 10.** Contingency table, column relative frequencies: where do you buy organic food the most? Respondent's country of origin.

|  | Czech Republic | Slovak Republic | Germany |
|---|---|---|---|
| Mall, supermarket | 26.64% | 31.00% | 39.57% |
| Health food and organic food stores | 18.35% | 13.33% | 12.82% |
| Farms, farmers markets | 7.33% | 6.00% | 18.55% |
| Smaller grocery stores | 4.43% | 5.67% | 2.56% |
| Pharmacies | 2.90% | 5.00% | 3.42% |
| I plant it myself | 14.29% | 14.00% | 4.27% |
| Other | 26.06% | 25.00% | 18.80% |

Source: (authors' calculations).

The majority of respondents who do not buy organic food at all may be found in the Slovak Republic, as opposed to Germany, as seen in Table 11. Outside of the Slovak Republic, there are significantly more respondents who buy organic produce several times a month. Once again, the country with the lowest number of respondents who shop for organic food several times a week, turned out to be Slovakia. The significant statistical dependence is borderline, the *p*-value is 0.05.

**Table 11.** Contingency table, column relative frequencies: how often do you buy organic food in your household? Respondent's country of origin.

|  | Czech Republic | Slovak Republic | Germany |
| --- | --- | --- | --- |
| Several times a week | 18.33% | 16.36% | 21.51% |
| Several times a month | 21.33% | 17.40% | 23.08% |
| About once a month | 13.67% | 14.62% | 15.38% |
| Less often | 17.67% | 18.19% | 18.66% |
| Not at all | 13.67% | 16.86% | 10.26% |
| No answers | 15.33% | 16.57% | 11.11% |

Source: (authors' calculations).

Organic food of foreign production is purchased in Slovakia slightly more often, while regional production is more popular in Germany, as seen in Table 12. The proportion of domestic production remains approximately the same in all three regions. The significant statistical dependence is borderline, the *p*-value is 0.06.

**Table 12.** Contingency table, column relative frequencies: In assortments of available organic products there prevail products of: Respondent's country of origin.

|  | Czech Republic | Slovak Republic | Germany |
| --- | --- | --- | --- |
| Domestic production | 33.47% | 32.67% | 32.48% |
| Foreign production | 9.15% | 14.00% | 11.11% |
| Regional production | 10.89% | 12.33% | 17.09% |
| I do not know | 29.37% | 25.67% | 28.21% |
| No answers | 17.11% | 15.33% | 11.11% |

Source: (authors' calculations).

The correspondence map (Figure 3) confirms that Czech respondents often purchase domestic products, while Slovak respondents prefer foreign produce.

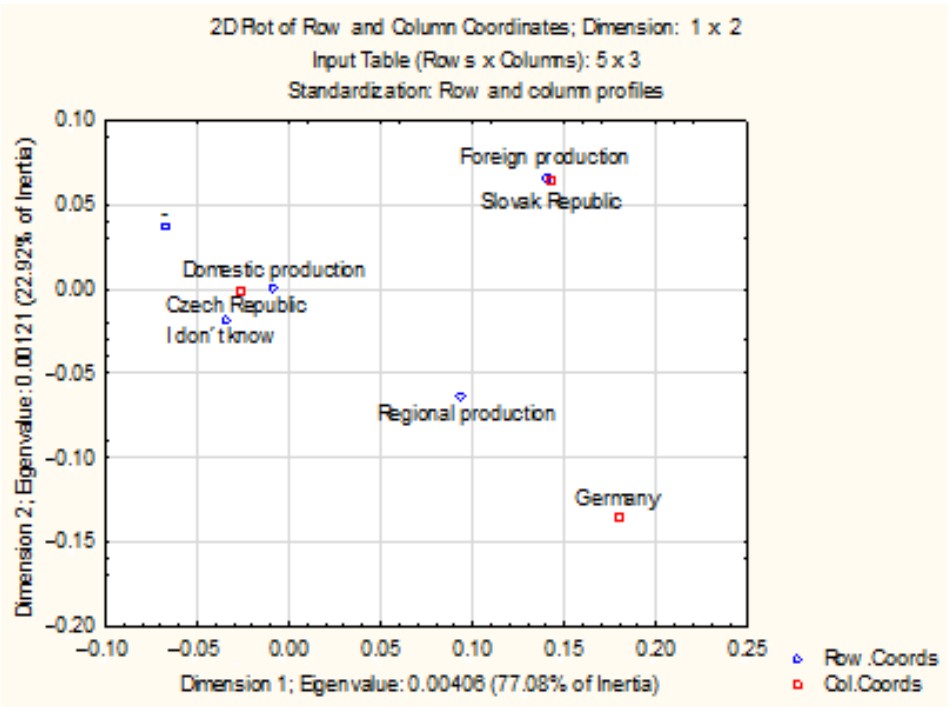

**Figure 3.** Correspondence map: in assortments of available organic products there prevail products of: respondent's country of origin, source: (authors' calculations).

The logistic regression method was used for a detailed comparison of the situation in the field of organic food in the selected countries. Using logistic regression, it is possible to

model the dependence of a binary variable (do you buy organic food in your household?) on several identification variables at the same time and thus compare consumer behaviour in the surveyed countries in different demographic categories of respondents. We examined the dependency of the frequency of purchasing organic food on sex, age, education, and monthly household income of the respondents. The variables of age, education, and monthly household income were scaled on a growing scale (1, 2, 3, . . . ). The variable of sex was scaled with 1 (male) and 0 (female).

The results of the logistic regression are shown in Table 13. All logistic regression models are statistically significant (likelihood ratio statistics, *p*-value < 0.001, goodness-of-fit test, *p*-value 1.000).

**Table 13.** Logistic regression coefficients.

| Country | Variables | Coefficient Beta | Wald Statistic | Significance |
|---|---|---|---|---|
| Czech Republic | Constant | 0.32 | 1.25 | 0.2644 |
| | Sex | −0.10 | 3.79 | 0.0314 |
| | Age | 0.05 | 0.72 | 0.4015 |
| | Monthly income | 0.25 | 8.15 | 0.0084 |
| | Education | 0.30 | 7.29 | 0.0069 |
| Slovak Republic | Constant | −1.09 | 23.03 | 0.0000 |
| | Sex | −0.10 | 5.10 | 0.0239 |
| | Age | 0.04 | 0.70 | 0.4015 |
| | Monthly income | 0.33 | 23.04 | 0.0000 |
| | Education | 0.32 | 14.83 | 0.0001 |
| Germany | Constant | −0.59 | 3.70 | 0.0544 |
| | Sex | −0.08 | 3.62 | 0.0431 |
| | Age | −0.05 | 0.49 | 0.4833 |
| | Monthly income | 0.06 | 1.73 | 0.1879 |
| | Education | 0.11 | 3.99 | 0.0319 |

In all the countries surveyed, a significant dependency of purchasing organic food on the level of education was established: the higher the level of education, the greater the proportion of consumers purchasing these products. This effect was the strongest in respondents from Slovakia, slightly weaker in Czech respondents, and the least in German respondents. The frequency of purchasing organic food strongly depends on the respondents' sex, with women shopping for organic products significantly more often. This factor was of least influence in Germany. With increasing income, the frequency of purchasing organic food grows significantly as well: the most in Slovaks, and less in Czechs. The increase in the frequency of purchasing organic food with more income is noticeable in German respondents as well; however, this dependency is not statistically significant. A significant dependency of purchasing organic food on age was not shown in any of the countries surveyed. Nevertheless, it is obvious from the regression coefficient values that the frequency of purchasing organic products slightly increases with age in the Czech Republic and Slovakia, whereas in Germany, it slightly decreases.

## 5. Discussion

This study searches to provide a comparison of customer preferences regarding organic food in different regions of the EU. Initial research performed by this team of authors previously highlighted several issues: organic produce in the Czech Republic is most frequently purchased by respondents in the age of 45+ years, who also tend to spend more money for this range of products [54]. With increasing age, the frequency of buying organic food increases. However, young respondents often grow their own organic produce. As it turns out, when it comes to organic products, Czech respondents purchase most frequently fruits and vegetables, milk, and dairy products. Zámková and Prokop [50] proved that the number of young Czech and Slovak respondents shopping for organic food regularly is

only negligible (less than 8%). The present research now suggests that the organic food purchase frequency is generally higher in the west of the EU, compared with the Czech and Slovak Republics. Organic food is purchased more often by women and respondents from households with a higher living standard [78]. Respondents tend to shop for organic food mainly in malls and supermarkets. Besides the primary reason for which consumers decide against buying organic food, its price, respondents also do not believe organic food to be better than conventional food. According to Zámková and Blašková [79], a considerable number of respondents do not trust in the benefits of organic food. Even though Organic farming can represent an alternative to the classic intensive production which is typical for the Czech Republic and Slovakia and can offer higher added value to the farmers [80,81]. These alternative approaches became topical concerning the impact of climate change on classic agricultural production [82,83]. Our present analysis revealed that this lack of trust is more serious in the Czech Republic and Slovakia, than in the west of the EU.

Moreover, it should be noted that other authors focused on the issue of organic farming. Ditlevsen et al. [18] identified increasing demand for organic food throughout the West, while Petrescu-Mag and Petrescu [84] concluded that there is a need to increase consumers' trust in the superior quality of organic produce. They even pondered the question, of whether it would not be more fitting to introduce more stringent European standards for organic products. According to our research, Germans are more convinced about the benefits of organic food than people in the Czech Republic and Slovakia. Pedersen et al. [25], proved that consumers prefer to purchase organic food produced in nearby countries, due to perceived negative implications of transport on the environment, trust in the given country, and the overall image of the country of origin. We did not identify significant differences in individual countries regarding the preference of the food's place of origin; almost 30% of European respondents do not check the origin of purchased food, which implies their lack of interest. Authors Wunderlich, Gatto, and Smoller [85] concluded that there were large levels of uncertainty about the perceived environmental impact of conventional foods and GMOs, and less uncertainty about organic food impact. Plus, consumers who believed organic food was beneficial were more willing to purchase it. The same applies for consumers who believed GMOs were harmful, they purchased less GMO food, but this correlation was weaker than in the case of organic food. The evidence that knowledge about environmentally friendly solutions affects shopping behaviours highlights the importance of consumer education, which can help eliminate the existing uncertainty. Additionally, our research showed that a large proportion of Czech and Slovak respondents is not convinced that organic food is free from chemicals and harmful substances, and such respondents therefore cannot believe that organic farming is eco-friendly. Jose and Koshy [86] analysed the factors influencing the consumers of organic food products; one of the strong factors is the concern about conventional food's safety, a driving force not only behind the positive attitude towards organic food, but also to an overall healthier lifestyle. Social pressure plays an even more important role. These factors are less significant in the Czech Republic and Slovakia once again due to the identified lack of trust regarding the safety and benefits of organic products. Janssen [44] used Structural Equation Modelling to illustrate that different attitudes towards organic food and actual purchases are influenced by the same determinants only with different relative importance. It emerged from the aforementioned research that the two most influential factors were naturalness, healthiness, and environmental protection, as well as the preference for local and domestic food, and the desire for enjoyment of eating, while higher price was a negative determinant. According to the work of Peschel, Orquin, and Mueller Loose [87] bigger and visually more salient labelling causes a considerable increase in consumer's attention and subsequently in his willingness to buy the product. Our research showed that the appearance of the product was more important for Czech respondents, while the brand played a more important role for the Slovaks. Prakash, Singh, and Yadav [88] employed Structural Equation Modelling in order to identify the important influencing factors in customer's decision-making process, including high quality consciousness, brand

consciousness, hedonistic shopping consciousness, price consciousness, brand loyalty, environmental consciousness, and health consciousness. Our research about organic food suggests that the high-quality consciousness is more dominant in the western member states of the EU. Pilar et al. [89] compared the results obtained from studies carried out using standard survey methods (questionnaires) with the results of a social network analysis based on Instagram posts. Both methods lead to the identification of the following key factors influencing organic food purchases: health consciousness, ecological motives, taste, and hedonism. The analysis carried out by our research team indicated that mainly citizens of Germany believed in the healthy and eco-friendly nature of organic food. The participants of a survey conducted by authors Faletar, Kovačic, and Cerjak [90] confirmed that the three most often bought organic vegetable species are: green salad, tomatoes, and carrots. The majority of respondents claimed to buy organic vegetable occasionally, and most often in specialized organic food stores. Sex, the number of household members, the presence of children in a household, self-rated knowledge about organic vegetables, all these factors have a strong influence on the frequency of purchases. Our research suggests that compared with Germans, Czech and Slovak customers tend to shop more often in specialized shops, as opposed to large stores, but the majority of grocery shopping still takes place in malls and supermarkets. Rani, Shah, Habib and Khan [91] studied the factors motivating the consumers in Pakistan to buy organic food, and they listed the following influencers: age, employment status, income, education, number of children, taste, no chemical residual, high nutritional value, freshness, and colour of vegetable, while incorrect product size and shape have an adverse effect on consumers. Apparently, food buying respondents in Pakistan prioritise their health. The households with higher income are aware of the negative impact on health caused by chemicals, and they are more likely to pay a high price for fresh organic vegetables. Our results show that respondents from the west of the EU seem to be more convinced about the health benefits of organic food. According to Von Meyer-Höfer, Nitzko, and Spiller [92], EU consumers generally expect organic food to be free of chemical pesticides and mineral fertilisers, they also expect the naturalness of organic food products and resource saving. However, further research showed, that some attributes (such as naturalness) might not mean the same to every consumer in every country and that can cause disappointment. These descriptors must be therefore used with great caution when promoting organic food. This ambiguity may be the reason behind the lack of trust in organic food discovered in our research. This goes hand in hand with the research by Hasimu, Marchesini, and Canavari [93]. They found out that not unlike developed countries, Chinese consumers considered organic produce to be healthier, safer, and more expensive than non-organic food. However, in China organic is not necessarily synonymous with natural, since the translation of "organic" implies technology advanced products. In China, organic food is often being confused with the "green food" designation. Vendors of organic products should thus ensure proper communication to avoid misunderstanding. As part of the research conducted by Keramitsoglou et al. [94], the authors included actual packages of two organic products in the paper-and-pencil version, and a video presentation of those packages in the on-line version. The respondents seemed to be more willing to buy the organic product when they were presented live than on video, but their results were not unequivocal. The idea of the possible substitution of actual goods by video presentations for marketing and advertising purposes requires further research. For example, in Slovakia, organic food is generally underpromoted, and almost the same applies to the Czech Republic. Among others, Braun, Rombach, Häring, and Bitsch [95] addressed the question of insufficient usage of organic produce. Even though organic food is generally important in school catering in Berlin, locally produced organic vegetables play only a minor role. The research identified the following reasons behind this: the lack of incentives for the use of locally produced organic food in the procurement guidelines, combined with a limited budget and no preprocessing facilities, despite the fact that school catering depends heavily on preprocessed food. The implementation of organic food in the Czech Republic or Slovakia would be surely accompanied by the same challenges, organic

produce usage in Czech and Slovak schools is today nearly non-existent. The research conducted by Cristache et al. [59] revealed the fact that the expansion of land dedicated to organic farming is accompanied by a certain decline in agricultural production; organic farming is therefore not likely to completely replace conventional farming.

## 6. Conclusions

This research explored consumer behaviour on the organic food markets in different regions of the EU, Czechia, Slovak Republic, and Germany. It focused on the differences in organic produce shopping strategies in the above-mentioned countries considering the issue of sustainable consumption.

The survey started with a focus on factors that may be perceived as less or more important when buying organic food. Apparently, the price represents a factor that bears the same importance for all respondents, regardless of their location. The quality of products is considered to be the most important by Germans. Quality is clearly a feature that remains more valued to the west of Central Europe. Respondents from the Czech Republic find the appearance more important. Brand and product availability, along with its place of origin seem to be the most important feature for Slovaks.

The next objective of the research was to determine respondents' attitude towards the quality of organic food. Germans reported more confidence in the healthier, higher quality, and more appealing nature of organic produce. Organic food is considered to be tastier more often by the citizens of the Slovak Republic. The overall results show that respondents from the Germany seem to be more convinced about the benefits of organic food, plus they claim it is recognizable in stores. The responses imply that organic produce is most accessible in the Germany.

From the marketing perspective, from the three respondent groups, the Slovaks reported organic food to be the least recognizable in stores, and the least promoted. Slovakia is also the country with the highest number of respondents, who have never seen any advertisement for organic food. This may represent an opportunity for marketing experts to change the stimulation of sustainable consumption. Our results show that all media outlets contribute almost equally to organic food awareness to stimulate consumers' behaviour towards a more sustainable buying behaviour in all of the countries.

Our data also suggest that approximately two-thirds of respondents agree with the support for organic farming coming from public sources, while Slovaks are the most accommodating. Germans tend to buy organic food regularly or more often, Slovak respondents are the least frequent to do so. Out of the three, the Slovak Republic is home to the highest number of respondents who do not buy organic food at all. Outside of the Slovak Republic, there are significantly more respondents who buy organic produce several times a month. This means that there is some room for improvement, when it comes to promotion of organic food mostly in the Slovak Republic. Big stores (malls and supermarkets) are most often frequented by respondents from Germany as opposed to the Czech respondents. Czech respondents tend to shop in specialized stores. German consumers prefer buying organic food on the farmers' markets. Only a very low number of Germans grow organic produce themselves, while it is more popular in Czech and Slovak republics. The research confirmed that Slovak respondents tend to buy more often foreign organic food, while Germans prefer regional produce. The proportion of domestic production remains approximately the same in all three regions.

The attitudes of respondents who do not buy organic food were also addressed. Those who do not buy organic food at all afterwards provided the reasons why. The most frequent reason was the price and lack of attractiveness, mainly in the case of Czech respondents. A rather large proportion of respondents is also suspicious of the fact, that organic produce does not contain any harmful chemical substances, mainly in the Czech and Slovak Republic. The same goes for the fact that respondents consider organic food to be an unnecessary luxury. In the Czech Republic, respondents struggle to find enough information about organic food and may complain of the small assortment of these products. To sum, the

research confirmed the prevailing distrust of Czech and Slovak citizens with regard to the benefits and qualities of organic food.

The research results lead to marketing recommendations for labelling coordinators in the individual countries, with these recommendations being different for the different countries as well. The results show that in Germany, labelling coordinators should focus their marketing activities and communication on campaigns mainly relaying factors such as the health benefits of organic food for customers, higher quality, and environmental friendliness of these products, and focusing on increasing solidarity with local producers of this food and purchasing directly from the farmers. The general awareness about the products labelled as "organic" is sufficient in Germany and as such, the current campaigns should mainly focus on the perception of the above-mentioned associations the customers have in relation to organic food based on the research results. The most significant customer group in Germany mainly involves women with higher education. In the Czech Republic, the coordinators of these labels should focus their marketing communication campaigns mainly on explaining the main benefits of organic food compared with conventional products, considering the fact that the related awareness is still very low, and the respondents often consider this food an unnecessary luxury, mainly due to its price. The availability and range of these products is still quite limited in the Czech Republic. The main target group is women with middle and higher income and higher education. In Slovakia, where the respondents' relationship to organic food is so far the least positive, the labelling coordinators should mainly focus their marketing communication on increasing awareness about organic food among consumers. The marketing communication campaigns should also target at supporting domestic (Slovak) organic food since the research results show that Slovak consumers usually choose foreign organic products. At the same time, the campaigns should communicate associations consumers in Slovakia should have in relation to the BIO label (health, quality, mainly egoistic reasons through rational marketing appeals) to eliminate the fact that products labelled as such are only seen as an unnecessary luxury compared with conventional food products. Based on the analyses carried out, the main target group is women with higher income and higher education.

Future research will focus on the impact of the COVID-19 pandemic on the organic food market and explore factors that may influence consumer purchasing behaviour, particularly the intention-behaviour gap or a green gap.

Limitations of the research: The main limitation of this study is for instance the fact that the authors abstract the division of shopping behaviour related to organic agricultural products for altruistic reasons. This division is yet another interesting aspect mainly for creating specific marketing campaigns, which are however not subject of this article and would be suitable for a marketing study, for instance for label coordinators in cooperation with the authors. Another limitation is that authors are mostly focused on the situation before the COVID-19 pandemic, which has influenced the market a lot. Therefore, the future research of the authors' team will be focused to determine the effects that the COVID-19 pandemic has had on the organic market in the selected countries, as the latest data signal that it has had a tremendous impact on people's purchasing behaviour. We expect that the respondents' awareness of organic, environmental and health issues has probably strengthened, the trend of online sales and subscription boxes for organic produce has grown, which is expected to have a positive effect on the sector.

**Author Contributions:** This article was written by European experts in economics and agriculture Conceptualization, S.R.; Data curation, M.Z. and L.P.; Formal analysis, M.Z. and M.C.; Funding acquisition, S.R., M.P. and K.M.; Investigation, M.C. and R.S.; Project administration, S.R., M.Z., L.P. and P.D.; Resources, M.P. and P.P.; Software, M.C., M.P., R.S. and S.N.K.A.-K.; Supervision, K.M., M.M., P.D. and P.P.; Validation, S.N.K.A.-K. and P.P.; Writing—original draft, S.R., M.Z., M.P. and M.C.; Writing—review & editing, S.N.K.A.-K. and M.M. All authors have read and agreed to the published version of the manuscript.

**Funding:** This research was founded by the CULS Prague, under Grant IGA PEF CZU (CULS) nr. 2019B0006, Atributy řízení alternativních business modelů v produkci potravin, and the Grant

College of Polytechnics in Jihlava nr. 1170/10/2136 Analysis of organic food purchase during the COVID-19 pandemic with using multidimensional statistical methods.

**Data Availability Statement:** Not applicable.

**Acknowledgments:** This research was supported by the CULS Prague, under Grant IGA PEF CZU (CULS) nr. 2019B0006, Atributy řízení alternativních business modelů v produkci potravin, and by the College of Polytechnics in Jihlava under Grant IGS nr. 1170/10/2136 Analysis of organic food purchase during the COVID-19 pandemic with using multidimensional statistical methods.

**Conflicts of Interest:** The authors declare no conflict of interest.

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
