# Peer review of "Pre-COVID-19 Organic Market in the European Union—Focus on the Czech, German, and Slovak Markets"

_agriculture, doi:10.3390/agriculture12010082_

Round 1

Reviewer 1 Report

Thank you for the opportunity to review this interesting article. However, we have made the following findings as follows:

1. Abstract. The authors clearly state the main objective of their research, indicating the results obtained from the investigation.

2. Introduction. The introduction is focused on the organic market in the 3 countries subject to research and is heading towards the end of the article. There is no section dedicated specifically to "Literature review" and what is presented is focused only on the 3 countries subject to research without reference to other countries in the EU or in the world. I suggest the authors introduce a subsection dedicated to the EU organic market.

3. Materials and methods. The research methodology is broadly described, but no information is given on the questions that were contained in the online questionnaire, nor is the percentage of respondents in each country surveyed to make a comparison of the final results obtained. Suggest that the authors clarify these issues.

4. The analysis of the results and the conclusions are eloquent and well outlined.

Author Response

Response to Editor and Reviewers

We would like to thank the Reviewer for their interest in our work and for their helpful comments that have greatly improved the manuscript. We did our best to respond to the points raised. The Reviewer brought up some good points, and we appreciate the opportunity to clarify our research objectives and results.

As indicated below, we considered all the concerns and specific comments provided by the Reviewer and made necessary changes accordingly.

We have changed the article due your recommendation. And we also checked and changed the English by a native speaker and style of the article.

Thank you for the opportunity to review this interesting article. However, we have made the following findings as follows:

  1. Abstract. The authors clearly state the main objective of their research, indicating the results obtained from the investigation.

Thank you for your comment.

  1. Introduction. The introduction is focused on the organic market in the 3 countries subject to research and is heading towards the end of the article. There is no section dedicated specifically to "Literature review" and what is presented is focused only on the 3 countries subject to research without reference to other countries in the EU or in the world. I suggest the authors introduce a subsection dedicated to the EU organic market.

We add more information about global and EU organic market to the Introduction part. We also divided the long Introduction part, extended „Literature review “par as the chapter 2.

  1. Materials and methods. The research methodology is broadly described, but no information is given on the questions that were contained in the online questionnaire, nor is the percentage of respondents in each country surveyed to make a comparison of the final results obtained. Suggest that the authors clarify these issues.

We add more information about our research aim and questions to the chapter 3. Materials and Methods.

  1. The analysis of the results and the conclusions are eloquent and well outlined.

Thank you for your comment.

Reviewer 2 Report

The introduction contains irrelevant information and there is a lack of relevant one.
Introduction is super long. In fact it is not any information and Why is relevant this study?. What is the
contribution? What is the state of art? How does it advance on the topic?.
Literature review: This section does not provide any relevant information. Please provide a literature review section with the most relevant findinds on the topic.

Your results are merely descriptive, authors should include more analysis. Thus, the findings are poor. Same with conclusions.Please deep on them and provide recommendations for practicioners, researchers. 

Author Response

Response to Editor and Reviewers

We would like to thank the Reviewer for their interest in our work and for their helpful comments that have greatly improved the manuscript. We did our best to respond to the points raised. The Reviewer brought up some good points, and we appreciate the opportunity to clarify our research objectives and results.

As indicated below, we considered all the concerns and specific comments provided by the Reviewer and made necessary changes accordingly.

We have changed the article due to your recommendation. And we also checked and changed the English by a native speaker and style of the article.

The introduction contains irrelevant information and there is a lack of relevant one.
The introduction is super long. In fact, it is not information and Why is relevant this study?. What is the
contribution? What is the state of art? How does it advance on the topic?.
Literature review: This section does not provide any relevant information. Please provide a literature review section with the most relevant findings on the topic.

We add more information about global and EU organic markets to the Introduction part. We also divided the long Introduction part into two parts – Introduction and Literature review. We extended the „Literature review“ part as Chapter 2 and added more relevant studies focused mostly on the organic market and also studies about the organic market influenced by Covid-19. We add circa ten other relevant studies focused on the problem of our article.

Your results are merely descriptive, authors should include more analysis. Thus, the findings are poor. Same with conclusions. Please deep on them and provide recommendations for practitioners, researchers. 

We used Logistic Regression to improve our article and the relevance of the conclusions of our research and in our article. We also add table 12 and recommendations for marketing communication activities for the coordinators of those brands to the conclusion part.

Reviewer 3 Report

For detailed comments see the attached file

Author Response

Response to Editor and Reviewers

We would like to thank the Reviewer for their interest in our work and for their helpful comments that have greatly improved the manuscript. We did our best to respond to the points raised. The Reviewer brought up some good points, and we appreciate the opportunity to clarify our research objectives and results.

As indicated below, we considered all the concerns and specific comments provided by the Reviewer and made necessary changes accordingly.

Abstract. I would recommend describing the study results more briefly and including information on the study sample and method in the abstract.

We have changed the abstract of the article due to your recommendation (added information about study sample and methods)

Introduction. I propose to refer to the more fundamental theories of demand factors and, on this basis, try to determine whether in the sample consumer behavior is indicative of a typical variation of demand to price or whether it is of an atypical nature such as for luxury or inferior goods.

We added more theoretical background to the introduction part (55-63)

Editing remarks. When you write that the results are presented in percentages in the table title, you don't have to put the % sign next to each value (this remark applies to all tables). In the head of the table in the first column, the variants of answers are presented, but not the year 2016 or 2019. So. please rename the head of the first column (Table 1-4, 8-13) and introduce the head (Table 5-7).

We have changed it due to your recommendation

We also checked and changed English by a native speaker and style of the article.

Round 2

Reviewer 2 Report

My concers have not been properly adressed. Introduction does not answer the relevance the paper, what is new and the rest of the questions. 

Literature review, please include a table with the previos research and findings, emphasising the value of previous research.

Please include a table with the profile of sample and explain how the survey was done. and participants contacted. Obviously, the countries are different and also the profile of citizents and consumer behaviour regarding organic products. Authors should not conclude that "Due to the similar numbers of respondents in all the countries and the use of a quota sample of respondents, the data from the individual countries are comparable." This assertion is wrong.

What is the meaning of the regresion model  on your study? What is its contribution? what is the significance¿ pleae provide the vakues of  beta and other relevant indicators.

Your results are not always supported by your data. For example, Authors say "The respondents aged 25 years and less tend to purchase organic produce least frequently, and they also often do not care about the origin of organic produce" but how many of the respondents are below 25? your data in sample say that more respondents under 35 years of age (63%)". Nothing about 25 years old.. Authors should revise carefully their results ams assure that results are based on their research

Author Response

Review round 2

We would like to thank again to the Reviewer for their comments and helpful recommendations on how to improve the article. We have changed the article due to your recommendations.

We modified the Introduction part, added more information about the organic market, and inserted into the Literature review part a Table “Common factors affect on the organic markets in selected countries „.

A table with the profile of the sample of respondents was inserted. The citation regarding the age of the respondents was adjusted. The other results were checked. The benefits and significance of the regression model have been added. Beta coefficients are given in Table 13. P-values of Goodness-of-Fit statistics were added to the comment to Table 13.

Added data and information are marked by a red color.
